# Relationship between Pelvic Dimensions and Maximum Traction Forces Required during Parturition in Holstein Cows Using a Biomechanical Obstetric Simulator

**DOI:** 10.3390/ani14132011

**Published:** 2024-07-08

**Authors:** Angeliki Tsaousioti, Anastasia Praxitelous, Michail Patsikas, Meik Becker, Heinrich Bollwein, Constantin M. Boscos, Evangelos Kiossis, Georgios Tsousis

**Affiliations:** 1Clinic of Farm Animals, School of Veterinary Medicine, Faculty of Health Sciences, Aristotle University of Thessaloniki, 54627 Thessaloniki, Greece; tsaoange@vet.auth.gr (A.T.); praxitea@vet.auth.gr (A.P.); pboscos@vet.auth.gr (C.M.B.); ekiossis@vet.auth.gr (E.K.); 2Laboratory of Diagnostic Imaging, School of Veterinary Medicine, Faculty of Health Sciences, Aristotle University of Thessaloniki, 54627 Thessaloniki, Greece; patsikm@vet.auth.gr; 3Clinic for Cattle, University of Veterinary Medicine Hannover, 30559 Hannover, Germany; meik.becker@tiho-hannover.de; 4Clinic of Reproductive Medicine, Vetsuisse Faculty, University of Zurich, 8057 Zurich, Switzerland; hbollwein@vetclinics.uzh.ch

**Keywords:** parturition, traction forces, dairy cattle, pelvic measurements

## Abstract

**Simple Summary:**

Parturition without complications is a prerequisite for a successful productive life in dairy cows. This study aimed to investigate the association between the pelvic dimensions of the dam with the maximum traction forces required at parturition using a biomechanical obstetric simulator. The effect of calf body dimensions was additionally investigated. An exponential relationship between maximum traction forces and pelvic dimensions was principally found, whereas the relationship with calves’ measurements was frequently linear and of lower magnitude. Pelvic dimensions affected the traction forces recorded at the entrance of the calf’s elbows and pelvis to a greater proportion, whereas during the entrance of the thorax, both the dam and calf dimensions proved equally important. Our results indicate that, regarding pelvic dimensions, critical cut-off points exist, below which a significant increase in the required traction forces is expected.

**Abstract:**

The primary aim of this study was to investigate the effect of the pelvic dimensions of Holstein cows on the traction forces during parturition. Additionally, the relationship between calf measurements and traction forces was explored. For this purpose, a modified in vitro biomechanical model simulating obstetric tractions was used. For the requirements of the experiment, six bone pelvises of deceased Holstein cows were collected based on their estimated pelvic inlet area (EPA) and prepared. Additionally, six stillborn calves were collected based on their body weight (BW). The parameters of the pelvic inlet and cavity were measured using computed tomography (CT). Using the simulator, every calf was pulled in a random order through all pelvises, realizing a total of 36 obstetrical tractions, and the required forces were documented with appropriate software. In each extraction, three peaks of forces were recorded, with the first peak occurring at the entrance of the elbows into the maternal pelvic cavity, the second peak at the entrance of the thorax, and the third at the entrance of the calf’s pelvis. Logistic regression revealed an exponential relationship between pelvic parameters and traction forces for the entrance of the elbows and the pelvis, with the recorded forces being higher in the two smallest pelvises and stabilizing at a lower level thereafter, while for the entrance of the thorax, the correlations were either exponential or linear. The adjusted coefficients of determination (r^2^) were generally above the threshold of 0.5 for the entrance of the elbows and pelvis and lower (0.3–0.4) regarding the thorax and were statistically significant (*p* < 0.05) in all cases. Regarding the relationships between the calf dimensions and the required traction forces, the types of correlations were primarily linear and of lower magnitude. The combination of pelvic and calf parameters in a multivariate model resulted in an r^2^ of 0.72 for the entrance of the elbows using the pelvic diagonal and calf’s body weight, an r^2^ of 0.62 using the pelvic area and calf’s thoracic circumference, and an r^2^ of 0.75 using the pelvic diagonal and calf’s fetlock joint width. In conclusion, under the conditions of the present experimentation, the applied traction forces were mainly influenced by the pelvic dimensions in an exponential manner, whereas the calf body measurements showed a weaker effect. Based on these findings, critical cut-off points exist, different for every pelvic parameter, below which a significant increase in the required traction forces is expected.

## 1. Introduction

The welfare of a dam and its productivity are largely dependent on the successful outcome of parturition. Based on estimations of veterinary surgeons, dystocia is one of the most painful events that a cow could experience [1,2]. Furthermore, dystocia has been linked to several pathological conditions affecting both the mother and the offspring. These include an increased risk of developing metritis and ketosis, as well as neonatal pneumonia and diarrhea. The severity of these consequences is further exacerbated as the dystocia score increases. It has been demonstrated that there is a correlation between the level of assistance provided during parturition and the involuntary culling of heifers and cows [3,4]. Similarly, the morbidity and mortality of calves were found to be positively correlated with the severity of dystocia [5], with mortality rates exceeding 50% in the most severe cases [6].

Primiparous cows are more prone to dystocia than multiparous cows, mainly due to the increased risk of fetomaternal disproportion [7,8]. Pelvic size, and in particular pelvic area (PA), is the second most important factor influencing dystocia incidence after calf birth weight (BW) [7,9]. In a recently published study by our working group [10], we found significant relationships between different pelvic dimensions and dystocia incidence in Holstein heifers and obtained critical cut-off points below which dystocia incidence significantly increased. Specifically, quartile analysis showed that heifers with a PA of less than 333.2 cm^2^ were almost three times more likely to experience dystocia than heifers with a larger inlet (19% vs. 7.7%), while several other pelvic parameters (hip width, pelvic volume, etc.) were similarly significant [10]. Furthermore, in the same study, regression analysis showed that the incidence of dystocia gradually increased as pelvic dimensions decreased, following an exponential function [10].

Regarding the effect of calf dimensions on dystocia, there are many studies supporting the significant positive association of dystocia incidence [7,11] and dystocia severity [12,13] with calf BW. In addition, the conformation of the calf contributes to dystocia as a causative factor due to fetomaternal disproportion [10,14,15,16]. Specifically, head circumference [14,17], thoracic circumference [10,14,18], and fetlock circumference and width [10,14] influence the incidence of dystocia. Becker et al. [19], using computed tomography (CT), found that the thoracic circumference in the region of the cranial sternum was the absolute largest circumference of the neonatal calf, while the width between the greater tubercles of the humeri and the trochanters of the femurs were the widest body widths and could play a critical role during parturition. However, the relationship between pelvic/neonatal dimensions and extraction forces during dystocia remains unclear.

Members of our research team have previously developed an in vitro biomechanical model that simulates the extraction of the bovine fetus and allows the objective measurement of the traction forces exerted [20]. Until recently, recommendations regarding the appropriate method of fetal extraction in bovine obstetrics were mainly based on practical experience [21,22] rather than scientific evidence [20,23]. Most of the published studies on labor progress and dystocia used a subjective method to estimate the degree of dystocia or different types of scoring scales due to the lack of an international dystocia classification system. This fact could cause bias in these studies because of the subjective and qualitative estimation of the procedure.

The aim of the present study was to increase the knowledge of the effect of pelvic and neonatal size on the extraction forces required during labor using an in vitro biomechanical model. Our main hypothesis was that the forces required to extract the calf using a standardized procedure would increase with progressively smaller pelvises. In addition, the contribution of calf size to the development of extraction forces was estimated.

## 2. Materials and Methods

### 2.1. Ethical Statement

This study was approved by the Research and Ethics Committee of the Faculty of Veterinary Medicine, Aristotle University of Thessaloniki (1182/8 March 2018). No operations on live animals were carried out. 

### 2.2. Study Design

For the purposes of this study, a modified in vitro biomechanical model that simulated the extraction of the bovine fetus from a bony pelvis was used. Six pelvic specimens were collected from Holstein cows and six stillbirth calves of the same breed. Pelvic measurements were performed with computed tomography (CT) after the completion of tractions. Calf measurements were taken using a caliper, tape measure, and electronic scale.

### 2.3. Eligibility and Preparation of the Bony Pelvises

Six bony pelvises from primiparous and multiparous Holstein cows (25–76 months old) that died or were euthanized for various reasons were collected and anatomically prepared for the experiment. Their eligibility was based on the estimated pelvic inlet area (EPA), which was calculated in cm^2^ using the following equation [24]:EPA= −349 + 6.5TcTc + 8.3TcTi − 1916/A
where TcTc is the most lateral point of the two tuber coxae (in cm), TcTi is the cranial point of the tuber coxae until the most caudal point of the ipsilateral tuber ischiadicum, and A is the age in months.

Measurements were taken on the deceased cow prior to collection using a custom-made caliper. Pelvises from cows that approximated the required dimensions based on previous studies [24] (350 to 500 cm^2^ in 25 cm increments) were eligible. After preparation, the pelvises were stored at −18 °C and placed at 4 °C 24 to 48 h prior to the start of each traction. Each pelvis was wrapped with a 4.5 × 0.1 cm self-adhesive rubber band to replace the support of the removed sacro-tuberal ligaments.

### 2.4. Eligibility and Preparation of the Calves

Six stillborn Holstein calves, with body weights (BWs) between 33.6 and 46.5 kg and no visible or palpable fractures, were used. The calves were collected immediately after death, cleaned, and stored at −18 °C. The calves were placed in a refrigerator at 4 °C 72 to 96 h before each traction. After thawing, the calves were weighed using an electronic scale, and specific body measurements were performed using a caliper and a measuring tape (Table 1). 

### 2.5. In Vitro Biomechanical Model

Becker et al. [20] developed and described an in vitro biomechanical model, which consists of two parts (Figure 1). Part A includes two supporting tables for calf movement and a pelvis-holding device that rotates freely in its transverse axis to adapt to calf movements during extraction. The pelvises are firmly fixed in the device using an adjustable hook and eye wire rope stainless-steel turnbuckles. During the experiment, the pelvises were placed in the left lateral position in the device, while the calves were positioned on the right side, facing the pelvic inlet in an anterior presentation and dorso-sacral position. Two obstetrical chains were placed on each metacarpus of the calf, above the fetlock joint, and were tied with the ropes of the traction unit of part B. Part B comprises the device that carries out the automated tractions and the force-measuring devices. The traction device has a control unit that allows the selection of pre-programmed traction modes and two independent electrical motors, each of which powers a horizontal threaded spindle with a loose flange. All tractions were performed at a constant velocity of 8 cm/min. The measuring device comprises two load cells interpolated between the loose flange and the ropes connected to the obstetrical chains. In this study, we used custom-made software (Vetlogger^®^ v. 1.0, Advantech SA, Glyfada, Greece) to convert the electrical signal from the load cells (S Type load cells, Sartorius SA, Hamburg, Deutschland) into a measurable force (Newtons). The software was installed on a computer running a Windows 10^®^ operating system. It recorded and stored the forces applied in digital format. The software enabled the continuous recording of the forces exerted on each limb, separately, during traction, with real-time force vs. time graph generation at 2 s intervals.

### 2.6. Extraction of the Calves

Each calf was extracted in a pre-programmed manner without manual intervention. Based on the results of Becker et al. [20], the actual experiment was preceded by an adjustment traction in simultaneous mode. Traction was then applied by alternately pulling the forelimbs 10 cm apart until the elbows entered the pelvic cavity, followed by simultaneous traction of both forelimbs until the calf’s pelvis completely exited the mother’s cavity. Each calf was extracted once through each of the six pelvises based on a random sequence provided by the Random Sequence Generator (Randomness and Integrity Services Ltd.^®^, Dublin, Ireland). After the generated pelvis was placed in the holding device, the calf was placed on the table in the right lateral, anterior presentation, and dorso-sacral positions. A lubricating gel (Vet Gel Lubricant^®^, Kerbl, Germany) was applied to both the calf and the pelvic cavity. A Krey and Schöttler double hook was placed on the lower jaw of the calf, and the two obstetrical chains were placed on each metacarpus. The calf was manually pulled into the pelvic cavity in a position where the forehead was caudal to the promontory of the sacrum and the carpal joints were below the lower jaw. A 5 kg weight was then connected by rope to the Krey and Schöttler hook to support the head in a horizontal position, and the birth chains were connected by rope to the load cells. Prior to the start of each pull, both forelimbs were placed at the same level, with the elbows of the calf positioned 5 cm caudal to the pubic tubercle. The pelvis was positioned at a 30-degree angle to the longitudinal axis of the traction direction. Traction was first applied manually until a steady tension of 4 kg was achieved on both forelimb ropes. The recording software was then initiated simultaneously with the alternate mode of traction. After the elbows fully entered the pelvic cavity, indicated by the head exiting and a decrease in traction forces, traction was briefly paused. The holding device was then stabilized to ensure the pelvis was parallel to the extraction direction, and the traction mode was changed to simultaneous. This was the only manual intervention during the extraction. Extraction continued without manual intervention until the calf’s pelvis had fully exited the dam’s pelvic cavity. The subsequent pelvis, determined by the random sequence, was then positioned in the device, and the entire procedure was repeated. A total of 36 extractions were measured and analyzed.

### 2.7. Pelvic Measurements Using Computed Tomography

Pelvic measurements were obtained using a spiral 16-section CT scanner (Optima CT520, GE Hangwei Medical Systems, Beijing, China) at the Diagnostic Imaging Laboratory of the School of Veterinary Medicine, Aristotle University of Thessaloniki after the completion of the extractions for technical reasons. The pelvis was positioned at the center of the tomography table, supported by foam materials, to ensure that the sacrum was horizontally aligned in the longitudinal axis, with the pelvic inlet facing the gantry. A topogram was created to define the tomography area, followed by the acquisition of spiral images that were 1.25 mm thick, using an exposure of 80 mA and 100 kV. The images were transferred to an image-archiving and -processing system (PACS), and measurements were taken using the dedicated software (AW VolumeShare 5 GE, v. 4.0; Appendix A). Table 2 presents the pelvic measurements and their corresponding anatomical boundaries. The pelvic inlet was manually outlined and automatically calculated using the AW VolumeShare 5 GE software (Appendix A). Each measurement was taken three times, and the mean was used for further analysis.

### 2.8. Statistical Analysis

The statistical analysis was conducted using the SAS^®^ OnDemand for Academics platform (SAS Institute, Cary, NC, USA). Univariate and multivariate polynomial logistic regression were used to analyze the relationship between pelvic and calf measurements and the maximum extraction forces per area (entrance of the shoulder, thorax, and pelvis). The strength of the association was evaluated using the adjusted coefficient of determination (r^2^). Paired *t*-tests were used to examine differences in the maximum force between two consecutive pelvises or calves. Analysis of variance components was performed using type III sum of squares. A significance level of *p* < 0.05 was employed.

## 3. Results

### 3.1. Descriptive Statistics

Table 3 and Table 4 display the measurements of the pelvises and calves, respectively.

Regarding the recorded traction forces, each traction displayed three force maxima (Figure 2), which is in line with previous research [20]. The first maximum was observed when the calf’s elbows entered the pelvic cavity, the second at the entrance of the chest, and the third at the calf’s pelvis. The traction forces were abruptly reduced once both elbows had fully entered, coinciding with the calf’s head exiting the pelvic cavity (Figure 2). The drop at the exit of the chest and pelvis appeared smoother. During the entrance of the elbows, the procedure was discontinued in three out of 36 tractions due to recorded forces exceeding the upper functional limit of the in vitro model (150 kg, equivalent to 1471 Newton). These cases involved the smallest pelvis (Pelvis1) and the three heaviest calves, which exceeded a body weight of 40 kg. Moreover, the pelvis of Calf4 was unable to exit from Pelvis2, which was the second smallest pelvis in terms of various dimensions in this experiment. A value of 1471 Newton was used for statistical analysis in all these cases. It is important to note that all analyses considered the sum of traction forces from both forelimbs.

### 3.2. Associations of Pelvic and Calf Measurements with Maximum Traction Forces

The associations between pelvic measurements and maximum traction forces during the extraction of the calves were statistically significant in all cases (*p* < 0.05; Table 5). These associations were mainly exponential during the entrance of the elbows and the pelvis of the calves (Table 5; Figure 3) and of higher magnitudes, with values mainly above 0.5 to 0.6. However, the associations observed during the entrance of the thorax exhibited either exponential (in the case of EPA, Dhpi, Dhmin, and Hc) or linear (in the case of PA, Dia, Hpi, and Hmin) patterns, with lower magnitudes and r^2^ values of ≤0.35 (see Figure 3).

Pairwise comparisons revealed a statistically significant (*p* < 0.05) reduction in the traction forces during the elbows’ entry between the smallest (348.4 cm^2^) and the second smallest (400.8 cm^2^) pelvis based on the PA (Appendix A). During the entry of the thorax, the reduction in the traction forces was significant between Pelvis2 and Pelvis3 (454.6 cm^2^; Appendix A), whereas during the entry of the calf pelvis, it was significant both between Pelvis1 and Pelvis2 as well as between Pelvis2 and Pelvis3 (Appendix A).

Regarding the effect of the calves’ measurements on the maximum traction forces, the associations were mainly linear and of moderate or weak magnitude (Table 6 and Figure 4). During the entrance of the calf pelvis into the pelvic cavity of the dam, only the FLJW showed a very weak association (r^2^ = 0.10).

During the entrance of the elbows into the maternal cavity, there was a significant increase in the traction forces between the calf that weighed 38.6 kg (Calf3) and that weighing 42.6 kg (Calf4; Appendix A). During the entrance of the chest, a significant increase was recorded between the calf with the fourth- (Calf1; 71 cm) and the fifth-greatest (Calf6; 74 cm) chest circumference (Appendix A).

Regarding the results of the multivariate analysis for the traction forces applied during the entrance of the elbows, the maximum r^2^ was estimated at 0.83, and the Dia, Hpi, PA, Hmin, and calf’s BW were all statistically significant (*p* < 0.05). The final statistical model, which contained one maternal and one variable from the offspring, included the Dia and calf’s BW with an r^2^ = 0.72 (Appendix A). The same analysis for the forces during the entrance of the chest revealed a maximum r^2^ of 0.69, where the maternal Dhpi, PA, Hmin, and calf’s BW and CC were statistically significant (PA, Hmin, and CC) or had a *p*-value below 0.10 (Dhpi and BW). The final model included the PA and CC with an r^2^ = 0.62 (Appendix A). For the entrance of the pelvis, the Dia and FLJW were significant, with an r^2^ = 0.75 (Appendix A). The analysis of the variance of the components demonstrated that the influence of the pelvic variables that remained in the final models was more important compared with that of the calf’s variables, especially regarding the forces during the entrance of the elbows and the pelvis (Appendix A). 

## 4. Discussion

The objective of the present study was to investigate the influence of the pelvic measurements of the dam on the applied forces during the extraction of the bovine fetus. In conjunction with the findings recently published in another manuscript [10], our research group sought to ascertain whether there are limits regarding pelvic dimensions below which traction forces (in vitro) and dystocia incidence (in vivo) significantly increase. This could assist farmers and veterinarians in dystocia prediction antepartum. The inclusion of the specific body dimensions of the calves enabled more precise results and avoided bias. For the present experimentation, only the bony part of the pelvis was used, which complicates direct comparisons with the applied extraction forces on living animals. During parturition, the presence of embryonic fluids and the soft tissues of the genitalia facilitate the extraction of the fetus. However, when dystocia occurs, particularly due to fetopelvic disproportion, the primary challenge is defined by the skeletal structures of the mother and the offspring. Furthermore, it is important to note that the results presented here pertain solely to the specific combination of pelvises and calves under consideration. Any potential addition of a heavier calf or a narrower pelvis would undoubtedly influence the outcomes. Nevertheless, this study was designed to collect representative specimens with dimensions that increased in size in a progressive manner. The selection of the pelvis was based on the estimated pelvic area, which was calculated using external pelvic measurements and the age of the cow [23]. In contrast, the calves were selected based on their actual BW. The pelvic measurements used for the analyses were performed with CT, which is the most accurate pelvimetry method [24]. Although there were differences between the EPA and the PA estimated by CT, leading to a different order of the obtained pelvises, the results derived from the statistical analysis were not substantially affected.

The pelvic inlet area is the most important maternal parameter influencing dystocia, particularly in cases of fetomaternal disproportion [7,19,25,26]. The present study demonstrated that the PA affected the traction forces regarding the entrance of all obstetrically relevant body regions of the calf. The critical point, below which the traction forces began to significantly increase, was 400 cm^2^, while the traction forces reached high levels (approximately or over 1000 Newton) when the PA approached 350 cm^2^. It is noteworthy that in the aforementioned in vivo study, dystocia incidence in Holstein heifers exhibited a notable increase from 5% when the EPA was approximately 400 cm^2^ to nearly 25% when the EPA approached 300 cm^2^ [10]. Regarding the other pelvic parameters, the pelvic diagonal diameter was also found to be a significant factor influencing traction forces. The length of the diagonal diameter was found to be greater than that of the narrowest horizontal diameter of the mid-pelvis. This justifies the general recommendation of obstetrical guidelines for a slight turning of the cranial part of the calf’s body for the easier passage of its pelvis through the maternal pelvis [22,24]. Although the caudal height was statistically significant regarding maximum traction forces, the adjusted r^2^ was low–moderate (0.15–0.54), indicating that the caudal part of the maternal pelvis did not play an important (or biological) role in the successful extraction of the fetus. This finding can be explained by the fact that the sacrococcygeal part of the spinal column is relatively flexible, particularly considering the influence of relaxin during parturition, which allows for a significant increase in the available space for fetal passage.

Regarding the influence of calf measurements on traction forces during the entrance of the elbows and the chest, although statistically significant associations were evident, the adjusted r^2^ values were relatively low (0.14–0.27). These relationships were mostly linear, indicating that as the calf dimensions increased, the traction forces increased in a linear manner (Figure 4, Appendix A). In our study, calves were collected based on their body weight, as this is the most common parameter in the current literature. We aimed to collect calves with representative and progressively increasing body weights for the breed. However, selecting calves with heavier body weights would have likely increased their importance regarding traction forces but would have also biased our results. It is also possible that the aforementioned linearity will change into an exponential function if calf dimensions exceed certain limits. A similar finding was observed in our study on dystocia prevalence [10]. It is generally accepted that body weight is the most important fetal factor that may cause dystocia [7,9]. Alongside other body measurements, such as chest circumference, which have been found to affect dystocia [10,16,18,27], these would influence the exerted traction forces. Furthermore, as is evident from Table 4, there is no consistent correlation between obstetrical relevant dimensions and body weight. For example, the largest chest circumference (79.5 cm) was observed in the third-heaviest calf (42.6 kg) in the present study. Despite the described moderate association, our results corroborate the hypothesis of Becker et al. [19], stating that the circumference of the chest is largest in newborns and may play an important role in obstetrics, as calves with a circumference greater than 74 cm required greater traction forces compared with calves with smaller chests (Appendix A). Another noteworthy finding of the present study is the minimal impact of calves’ pelvic dimensions on the maximum traction forces during the extraction of their pelvises. In the Holstein breed, hip-lock is a rare occurrence, and it is likely that it is primarily associated with the dam, in contrast with other breeds such as Belgian Blues, which experience a much higher incidence rate [16].

The analysis of the combined effect of pelvic and calf dimensions revealed that, under the conditions of the present experimentation, the effect of the maternal side was greater than that of the offspring. The combination of these variables in multivariate models could explain a significant proportion (up to 83%) of the variance in traction forces. A previous study conducted by our research group [23] demonstrated that the calf exerts a greater influence on traction forces than the traction method when a single pelvis is employed. Consequently, it is imperative that proper management of reproduction in dairy heifers is implemented, ensuring adequate body growth at insemination and the use of semen for ease of birth. Furthermore, the utilization of a competent obstetrical intervention when necessary is essential for a successful parturition without adverse effects for the mother and the offspring. 

## 5. Conclusions

The hypothesis of the present study was that the necessary forces for the extraction of a calf from a biomechanical model simulating parturition would increase as pelvic dimensions decreased. This hypothesis was validated. Furthermore, we found that these associations were mainly exponential, indicating that there are cut-off points, different for every pelvic dimension, below which traction forces significantly increase. Regarding the pelvic inlet area, which can be estimated using external pelvimetry and specific equations, this critical point is below 400 cm^2^. The findings of the present study could be used by herdsmen, farm staff, and veterinarians for the timely prediction of cows with a higher probability of requiring significant obstetrical assistance and for better calving management. 

## Figures and Tables

**Figure 1 animals-14-02011-f001:**
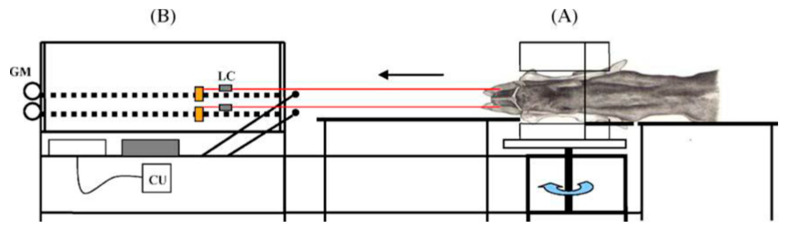
Schematic representation of the in vitro model viewed from the side: (**A**) = accommodation tables and holding device for the pelvis; (**B**) = traction unit with control unit (CU), gear motors (GMs), and load cells (LCs) to measure traction forces (adapted from Becker et al., 2010 [20]).

**Figure 2 animals-14-02011-f002:**
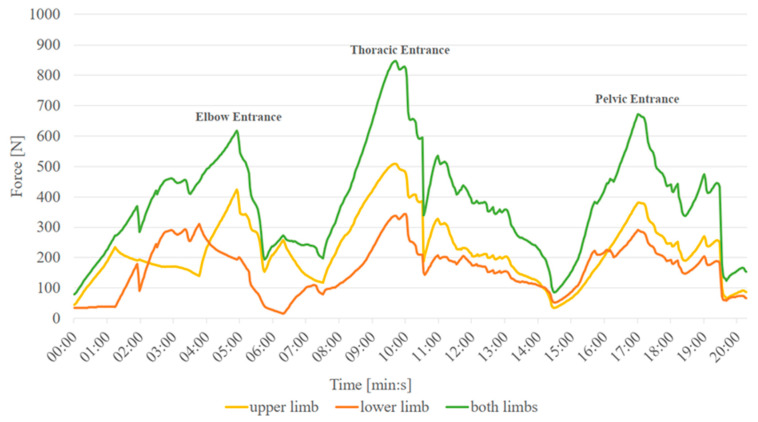
Model traction of Calf3 (38.6 kg body weight) through Pelvis1 (348.4 cm^2^ pelvic inlet area). The maximum values of the traction forces at the entrance of the elbows, the thorax, and the pelvis of the calf are denoted.

**Figure 3 animals-14-02011-f003:**
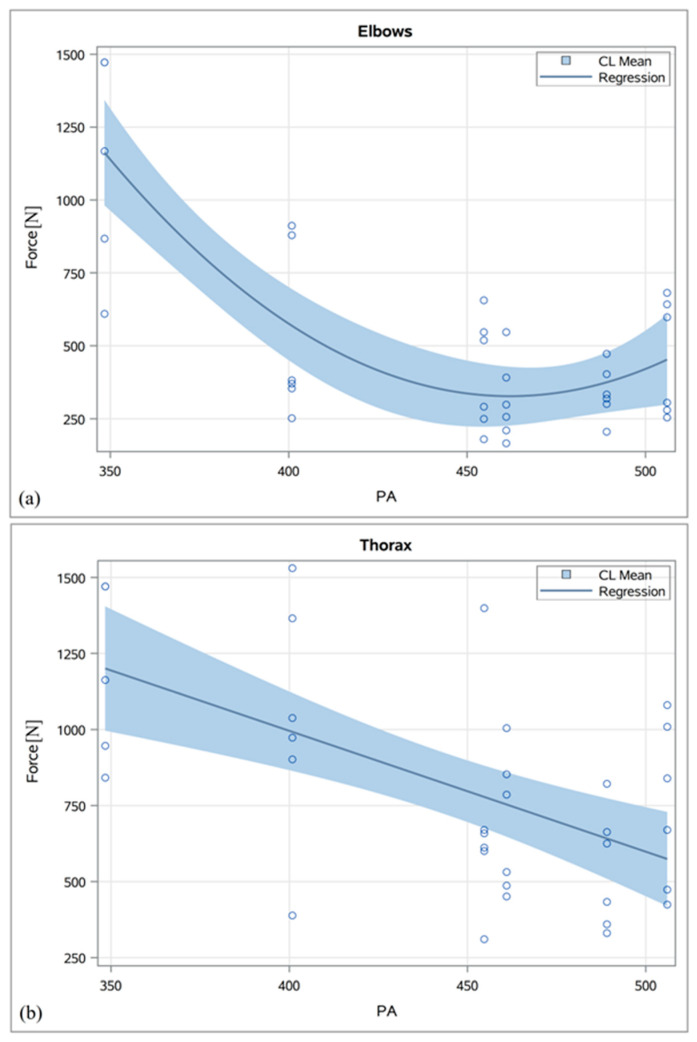
Predicted probabilities (with 95% CI) for maximum traction forces during the entrance of the elbows (**a**), the thorax (**b**), and the pelvis (**c**) of the calf dependent on maternal pelvic inlet area (PA).

**Figure 4 animals-14-02011-f004:**
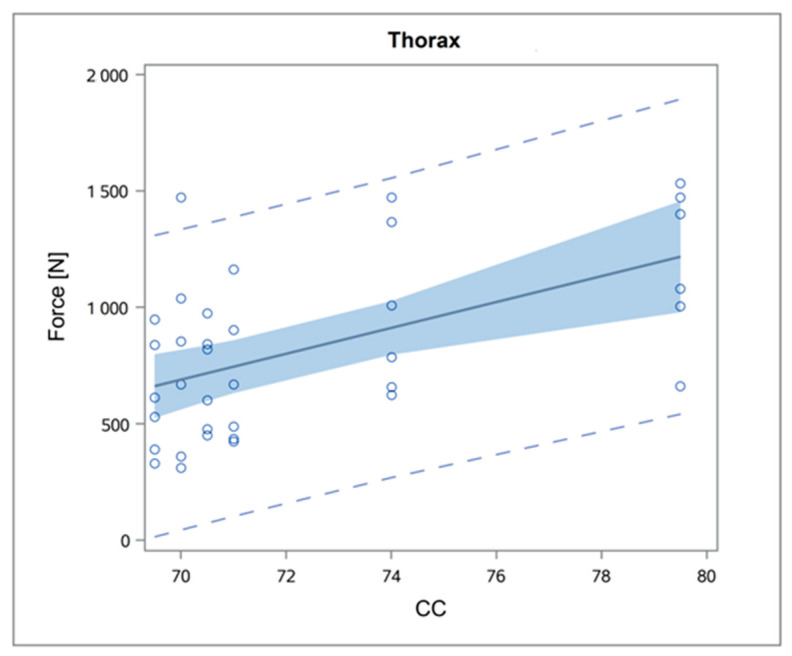
Predicted probabilities (with 95% CI) for maximum traction forces during chest entrance based on calf’s chest circumference (CC).

**Table 1 animals-14-02011-t001:** Body measurements of the calves with abbreviations (Abbr.) and definitions.

Body Measurement	Abbr.	Unit	Anatomic Borders
Body weight	BW	kg	Body weight
Head circumference	HCi	cm	Maximum circumference of the head on the level of orbitae
Fetlock joint circumference	FJCi	cm	Maximum circumference in the middle of the fetlock joint of the right forelimb
Fetlock joint width	FJWi	cm	Maximum width of the fetlock joint of the forelimb
Chest circumference	CC	cm	Maximum circumference in the region of cranial sternum
Hip width	TcTcC	cm	Most lateral point of the two tuber coxae
Pin bones width	TiTiC	cm	Most lateral point of the two tuber ischiadici

**Table 2 animals-14-02011-t002:** Pelvic measurements with abbreviations (Abbr.) and definitions.

Pelvic Measurement	Abbr.	Unit	Anatomic Borders
Estimated pelvic area	EPA	cm^2^	
Pelvic area	PA	cm^2^	
Medial horizontal diameter of pelvic inlet	Dhpi	cm	The two tuberculi musculi psoas minoris
Diagonal diameter of pelvic inlet	Dia	cm	Bottom edge of the os ilium at the point of the sacroiliac joint and upper edge of the contralateral eminentia iliopubica
Height of pelvic inlet	Hpi	cm	Promontorium ossis sacri to the most dorsal point of the tuberculum pubicum
Minimum height of pelvic cavity	Hmin	cm	Minimum distance between the most dorsal point of the tuberculum pubicum and the ventral area of the os sacrum
Narrowest horizontal diameter of mid-pelvis	Dhmin	cm	Smallest distance of the spina ischiadica
Caudal height of pelvic cavity	Hc	cm	Caudal end of the symphysis pelvina and the most caudal end of the ventral area of the os sacrum

**Table 3 animals-14-02011-t003:** Pelvic dimensions measured using computed tomography (except for estimated pelvic area (EPA)) classified by pelvic inlet area (PA).

Variable	Unit	Pelvis1	Pelvis2	Pelvis3	Pelvis4	Pelvis5	Pelvis6
PA	cm^2^	348.4	400.8	454.6	461.0	489.1	506.1
EPA	cm^2^	338.1	373.7	498.1	401.1	448.5	517.2
Dhpi	cm	17.5	18.9	22.6	20.4	22.4	23.2
Dia	cm	20.1	23.1	25.1	25.2	27.1	26.3
Hpi	cm	24.4	25.7	24.7	25.2	27.1	26.6
Hmin	cm	18.9	20.1	20.2	20.9	21.0	21.8
Dhmin	cm	16.5	18.4	20.2	18.5	19.1	21.2
Hc	cm	15.9	20.5	21.8	19.7	18.3	19.5

PA: pelvic area; EPA: estimated pelvic area; Dhpi: medial horizontal diameter of pelvic inlet; Dia: diagonal diameter of pelvic inlet; Hpi: height of pelvic inlet; Hmin: minimum height of pelvic cavity; Dhmin: narrowest horizontal diameter of mid-pelvis; Hc: caudal height.

**Table 4 animals-14-02011-t004:** Dimensions of the calves classified by body weight (BW).

Variable	Unit	Calf1	Calf2	Calf3	Calf4	Calf5	Calf6
Sex		F	M	M	M	F	F
BW	kg	34.7	37.7	38.6	42.6	43.5	46.5
HCi	cm	43	46	48	51.5	49	53
FLJCi	cm	15.5	16.5	16.5	14.5	17.5	17.5
FLJWi	cm	4.9	5.1	5.6	4.5	5.4	5.5
CC	cm	71	69.5	70.5	79.5	70	74
TcTcC	cm	14.2	18.8	16.4	15.9	20	15.2
TiTiC	cm	9.1	9.2	10.8	11.6	11	10.4

F: female; M: male; BW: body weight; HCi: head circumference; FLJCi: fetlock joint circumference; FLJWi: fetlock joint width; CC: chest circumference; TcTcC: hip width; TiTiC: pin bones width.

**Table 5 animals-14-02011-t005:** Adjusted coefficient of determination for the association between pelvic dimensions and maximum traction forces.

Variable	Elbows	Thorax	Pelvis
PA	0.63	0.33	0.66
EPA	0.58	0.35	0.62
Dhpi	0.63	0.35	0.65
Dia	0.62	0.37	0.64
Hpi	0.29	0.13	0.23
Hmin	0.63	0.28	0.61
Dhmin	0.60	0.27	0.56
Hc	0.54	0.15	0.46

PA: pelvic area; EPA: estimated pelvic area; Dhpi: medial horizontal diameter of pelvic inlet; Dia: diagonal diameter of pelvic inlet; Hpi: height of pelvic inlet; Hmin: minimum height of pelvic cavity; Dhmin: narrowest horizontal diameter of mid-pelvis; Hc: caudal height. Underlining indicates exponential association; no underlining indicates linear association.

**Table 6 animals-14-02011-t006:** Adjusted coefficient of determination for the association between calves’ dimensions and maximum traction forces during the extraction of 6 calves.

Variable	Elbows	Chest	Pelvis
BW	0.18	0.13	NS
HC	0.12	0.18	NS
CC	NS	0.27	NS
FLJC	0.18	NS	NS
FLJW	NS	0.16	0.10
TcTcC	NS	NS	NS
TiTiC	NS	0.14	NS

BW: body weight; HC: head circumference; CC: chest circumference; FLJC: fetlock joint circumference; FLJW: fetlock joint width; TcTcC: hip width; TiTiC: pin bones width; NS: not significant. Underlining indicates exponential association; no underlining indicates linear association.

## Data Availability

The data are contained within this article. The data are available from the authors upon request.

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
