# Peer review of "Relationship between Pelvic Dimensions and Maximum Traction Forces Required during Parturition in Holstein Cows Using a Biomechanical Obstetric Simulator"

_animals, 2024, doi:10.3390/ani14132011_

Round 1

Reviewer 1 Report

Comments and Suggestions for Authors

Dear authors,
Thank you for submitting this paper. As Becker et al. published the
first results by using this traction unit more than 10 years ago, this
topic is still relevant for everyone who is working with cows.
To be honest, it is not quit easy to understand how that meassuring
device works in detail, but you tried your very best to describe it
and you got good results. But it will not be easy to transfer them
into the field, as we are not able to use „CT-eyes“ predicting the PA.
So, please keep on doing research on that topic at least to save the
excellent work of Meik Becker.
Thank you
I had two small comments:
Author listing: H. Bollwein is marked with a 3, but he is working at
the University of Zuerich marked with a 4.
248 and 252: Does the superscription of Figure 2 exsit twice?
Under and over the figure?

Author Response

Dear reviewer,

We are very grateful for your motivating comments. I am personally very moved that you found the initial work of Meik excellent, as it was the first work I supervised and for other personal reasons I could explain in a personal communication if possible. We are completely aware that our results cannot be transferred per se in the praxis. However, taking into account also our other work regarding pelvimetry that we newly published, I think we have added a small piece of scientific evidence in the thematic.

We will try our best to keep the work in an area with such relevance for the praxis and the welfare of the cows, we all so much love.

Best regards,

Georgios

COMM1: Author listing: H. Bollwein is marked with a 3, but he is working at the University of Zuerich marked with a 4.

RESP1: Changed. Thank you for noticing.

COMM2: 248 and 252: Does the superscription of Figure 2 exsit twice? Under and over the figure?

RESP2: Probably we have had it initially over the figure, which is incorrect, and the editing of animals has placed it under as well. We have omitted the wrong placement.

Reviewer 2 Report

Comments and Suggestions for Authors

GENERAL COMMENTS

The manuscript reports the results of an interesting study exploring the relations of neonatal size and pelvic size on extraction forces during parturition in Holstein cows using a biomechanical simulation model. The materials and methods section covers all the complex activities involving the study, as well as the discussion part.

The study is, in my opinion, very stimulating, very articulated, and meticulous, and I think it worth the publication in Animals, after minor corrections regarding the explanation of results.

SPECIFIC COMMENTS

-Maybe the tabulation of uni-multivariate logistic regression should be disclose more in detail (perhaps in supplementary material).

-It would be interesting to report the equations of traction forces reported in Figures S1-S5, accompained by the statistical significance of function parameters.

I have 

Author Response

Dear reviewer,

We sincerely thank you for the very motivating comments. It is not common to get such nice feedback from a reviewer.

COMM1: Maybe the tabulation of uni-multivariate logistic regression should be disclose more in detail (perhaps in supplementary material).

RESP1: We are not certain as to which information from the uni- and multivariate logistic regression would be of interest. As we had many analyses due to the characteristics of the study, we decided to include the more meaningful ones, i.e. the results of the multivariate analysis. Please check the ones provided in the supplementary file and if necessary we can add more information in another round.

COMM2: It would be interesting to report the equations of traction forces reported in Figures S1-S5, accompained by the statistical significance of function parameters.

RESP2: This information has been added as a legend directly on the diagrams.

Reviewer 3 Report

Comments and Suggestions for Authors

I have had a great pleasure to read the presented work.

I have only few comments :

13 Who is working at the  Clinic of Reproductive Medicine, Vetsuisse Faculty ?

54 congratulations for this very nice introduction : consequences and influencing factors have been very well synthetized

122 months : a dot is lacking

172 do you think that a posterior presentation could have different results ?

193 can you add a CT picture to illustrate your measurements ?

217 congratulations forthe very nice presentation of the detailed results

341 what could be the force developped in case of a traction done by one or two human ?

399 can you add a comment on the method to evaluate the pelvic inlet area ?

Author Response

Dear reviewer,

We are very pleased that you enjoyed our work. Thank you for your time and efforts to improve our manuscript.

Best regards,

Georgios

COMM1: 13 Who is working at the Clinic of Reproductive Medicine, Vetsuisse Faculty ?

RESP1: Changed. Thank you

COMM2: 54 congratulations for this very nice introduction: consequences and influencing factors have been very well synthetized

RESP2: We sincerely thank you for the motivating comments!

COMM3: 122 months: a dot is lacking

RESP3: Added

COMM4: 172 do you think that a posterior presentation could have different results?

RESP4: I am convinced that a posterior presentation will have different results, as the sequence of the body parts of the calf is reversed. For instance, in posterior presentation we have the exit of the elbows first and then follows the head. That will change radically the development of the forces. It is funny, I work with this simulator for more than a decade and never occurred to me to try this! Thank you for brainstorming. 

COMM5: 193 can you add a CT picture to illustrate your measurements?

RESP5: For more clarity, we have provided all available CT measurements in the supplementary file.

COMM6: 217 congratulations for the very nice presentation of the detailed results

RESP6: Thank you very much once again!

COMM7: 341 what could be the force developed in case of a traction done by one or two humans?

RESP7: I actually use our simulator to teach students in cow obstetrics. I exclude the traction machine, and instead two students pull the calf. The forces we record with the load cells reach usually 40 to 50 kg per person, and only very strong students can get peaks of 70 to 80 kg. In this sense, our machine is actually pulling a bit over what actual people do. But it was constructed based on literature evidence from obstetric manuscripts.

COMM8: 399 can you add a comment on the method to evaluate the pelvic inlet area ?

RESP8: A comment has been added in line 400